# Quantitative evaluation of China's private universities provincial public funding policies based on the PMC-Index model

**Maobo Hu**[☯], **Cai Guo**[iD][☯], **Yang Wang**[☯], **Dan Ma**\*

Normal School of Vocational Techniques, Hubei University of Technology, Wuhan, Hubei, China

☯ These authors contributed equally to this work.
\* madan@hbut.edu.cn

**Data Availability Statement:** All relevant data are within the paper.

**Funding:** The funding was provided by National Social Science Fund (BIA190178) 'Research on

## Abstract

The special public funding policies, formulated and implemented by provincial governments, plays an important role in the development of private universities in China. However, there is a lack of scientific evaluation on the rationality and completeness of the provincial special public funding policy of China's private colleges and universities. Therefore, this paper uses PMC-index model and text mining technology to establish an evaluation index system for the provincial special public funding policy of private universities in China. Based on PMC-Index and PMC-Surface, 13 policy texts issued since 2010 were evaluated scientifically. The results show that the average PMC-Index of the 13 policies is 6.97, and the PMC-Surface map is smooth, which indicates that the overall structure of these policies is reasonable and the policy content has certain completeness. Only one policy is unacceptable. Through further analysis, we found that there is still much room for improvement in the content design of the sample policy, and its rationality and completeness are related to the publication time. This study is helpful to fully understand the advantages and disadvantages of the provincial special public funding policy of private universities in China.

## Introduction

Policy evaluation is an important part in the process of public policy analysis [1]. It can comprehensively evaluate the content, effect and value of policies [2, 3] by using a series of evaluation tools and methods [4], so as to provide information and basis for the continuation, amendment, termination and re-formulation of policies [5, 6].

Private higher education, as a "crowded quasi-public good" [7, 8], should be jointly provided by the government and the market, and its cost should be shared by the government finance and the educated [9, 10]. Many countries in the world have formulated and promulgated many public funding policies for private colleges and universities [11, 12], whose purpose is to promote the sustainable development of private higher education [13], improve the enrollment rate and completion rate of private university students [14], maintain educational equity [15] and so on. With the support of relevant policies, public funding has become an

public funding model and performance evaluation of non-profit private colleges and universities'. The funder was responsible for research design and manuscript writing in this study.

**Competing interests:** The authors have declared that no competing interests exist.

important source of funding for private higher education institutions in various countries [16]. In China, government financial support for private colleges and universities has almost become the consensus of all sectors of society [17], and different degrees of policy practices have been carried out in various regions. However, according to the effect of the policy implementation, the overall effect of the existing public funding policies is limited, and the funding level is obviously low [18]. At present, China's private higher education has entered a new era characterized by classified management, standardized promotion and connotation development [19], and the public funding system has entered a new stage of differential support [20]. This has brought great changes to the formulation of public funding policies for private colleges and universities [21]. In order to further improve the effectiveness of public funding, it is necessary to conduct a scientific evaluation of the public funding policy of China's private colleges and universities.

In fact, due to the large gap between policy goals and actual results, more and more scholars have conducted evaluation and research on the public funding policy of China's private colleges and universities. However, most of the existing studies focus on the qualitative evaluation of macro-level public funding policies, and pay less attention to the advantages and disadvantages of micro-level special public funding policies. At the same time, most relevant studies focus on the effectiveness of policies from the perspective of ex-post evaluation, but pay less attention to the integrity and rationality of policies themselves. This paper aims to bridge the gap of the above discussion and quantitatively evaluate the text of the China's private universities provincial public funding policies from a micro perspective, in order to provide a new basis for policy adjustment.

The main contributions of this study are as follows. First, it uses text mining technology to dig deeper into the policy of provincial special public funding for private colleges and universities in China, and evaluate the specific content of the policy, which provides a new perspective for related research. Second, it constructs the quantitative evaluation framework of the provincial special public funding policy of China's private universities based on the Policy Modeling Research Consistency (PMC) index model, which provides a new method for the existing research. Third, it selects 13 policies as research samples and makes a quantitative evaluation of the advantages and disadvantages of each policy, which provides new evidence for the adjustment of provincial special public funding policies for private colleges and universities in China. The rest of this article is structured as follows. Section 2 reviews the relevant literature. Section 3 introduces policy samples and methods. Section 4 presents and analyzes the empirical results. Section 5 elaborates the conclusions and shortcomings of the study.

## Literature review

The evaluation research on the public funding policy of China's private universities can be divided into the following two categories. The first is to evaluate the evolution, content and scientificity of relevant policies from the perspective of the policy itself. This kind of research mainly focuses on qualitative analysis, and quantitative research is rare. For example, Li [22] and other researchers conducted a holistic evaluation of the development of public funding policies in China's private colleges and universities, and explored the evolution trend of public funding policies in China's private colleges and universities by revealing the background, environment and significance contained in the process of policy evolution. Tan [23]; Huang et al. [24] adopted comparative analysis and content analysis method to study the content of public funding policies of private colleges and universities in China and abroad, and evaluated the funding models and funding intensity and other attributes of the connotation of policy texts, so as to provide beneficial references for improving the public funding policies of private

colleges and universities in China. The second is to investigate and evaluate the effect of policy from the perspective of policy implementation. This kind of research emphasizes the use of empirical means. For example, Bao et al. [25] introduced the propensity score matching(PSM) method to evaluate the comprehensive effectiveness of China's private higher education public funding policies from an empirical perspective, and found that although the public funding policies promoted private colleges and universities to strengthen student assistance, they failed to achieve the expected results in terms of reducing tuition burden, increasing teachers' salaries, and forming a crowding-in effect on private investment.

To sum up, the existing research has made a significant contribution to the improvement and optimization of the public funding policies of private universities in China. However, most of the relevant researches stay on the qualitative evaluation of macro public funding policies, and lack the systematic evaluation of micro specific special public funding policies. At the same time, the existing quantitative evaluation of relevant policies focuses on the study of policy effect, and the quantitative analysis of the completeness and rationality of the policy content itself is less.

At present, the methods commonly used in policy content evaluation include text analysis, mechanical element analysis, PMC-Index model and so on [26]. The text mining method mainly carries out simple word frequency statistics for policy topics, while the mechanical element analysis method needs to carry out subjective differential assignment for each dimension. Compared with them, PMC-Index model evaluation method has the following advantages: First, parameter setting is based on policy text mining, which reduces and avoids the subjectivity of human evaluation and can greatly improve the accuracy of quantitative evaluation. Second, not deliberately distinguish the importance of indicators at all levels, not set limits on the number and weight of variables, which is conducive to capturing all possible influencing factors; Third, it is convenient to analyze the consistency between policies and show the advantages and disadvantages of policies intuitively, so as to provide a more scientific and effective basis for policy improvement. Fourth, the cost is lower and easier to operate [27]. In recent years, PMC-Index model has been widely applied in the evaluation of public policy texts such as cultivated land protection policy [28], green development policy [29], waste separation management policy [30] and COVID-19 policy [31]. The above advantages indicate that the application of PMC index model to the evaluation of educational financial policy is scientific, rational and feasible. This paper tries to expand the application scope of PMC-index model, and uses it to evaluate the completeness and consistency of the provincial special public funding policy of private universities in China.

## Samples and methods

### Overview of China's private higher education public funding policy

The policy of private education is the decisive factor to promote the practice of private education in China [32]. Since the reform and opening up, China's private higher education public funding policies has undergone great changes, which has strongly supported the healthy and sustainable development of private higher education [33, 34]. In 1982, *the Constitution of the People's Republic of China* put forward the principle of "walking on two legs" in running a school, which clarified the legal status of private colleges and universities, and the Chinese government began to attach importance to public funding to private colleges and universities. However, it was not until 1997 that China's public funding to private colleges and universities was gradually institutionalized and standardized. In 1997, the State Council (SC) of PRC Issued *the Regulations on Running Schools by Social Forces*, requiring the government to provide financial aid to schools established by social forces in terms of land and tax revenue. In

2002, *The Law of the People's Republic of China on the Promotion of Private Education* was approved by the 9th Standing Committee of the National People's Congress (NPC), which provided a legal basis for financial support for private colleges and universities. In addition, the law states that the government can set up special funds to support the development of private colleges and universities. Since then, with the continuous reform and improvement of the economic system and the management system of private education, China's public funding to private colleges and universities has become diversified and scientific. In 2010, *the Outline of the National Medium and Long-term Education Reform and Development Plan (2010–2020)* proposed to actively explore the classified management of for-profit and non-profit private colleges and universities, and improve the support policy of public finance for private higher education. In 2016, *the Law of the People's Republic of China on the Promotion of Private Education* was revised for the second time, which provided the basis for the government to support and regulate the development of private higher education in the new era [35]. Furthermore, Its supporting document *Several Opinions of The State Council on Encouraging Social forces to set up Education and Promote the Healthy Development of Private education* proposes "implementing differentiated support policies" and puts forward new requirements for financial classification support for for-profit and non-profit private colleges and universities [36].

At present, China's private higher education public funding continues to grow, and has formed a funding policy model featured as rewarding excellence and supplemented by universal welfare. The content of the subsidy includes the funding support of private colleges and universities, tax incentives, credit incentives, land concessions and many other aspects. Among them, funds are the lifeblood of the development of private colleges and universities. The special public funding policy of private colleges and universities transformed the government's financial support behavior from policy support to both policy support and fund support [37]. In China, provincial governments are generally responsible for the financial support of private universities. Special funds for private colleges and universities refer to the funds allocated to private colleges and universities by provincial government departments for special purposes and for the completion of designated matters [38]. The provincial special public funding policy of China's private colleges and universities is a guiding document for local governments to support private colleges and universities with provincial special funds.

### Data sources and research samples

Considering that *the Outline of the National Medium and Long-term Education Reform and Development Plan (2010–2020)* proposed in 2010 to "actively explore the classified management of for-profit and non-profit private schools", most local governments' policies on private colleges and universities began to transform from "standardized management" to "financial support". This paper takes 2010 as the starting point of sample selection. In order to ensure the comprehensiveness of the policy samples, this paper takes "special funds", "special subsidies", "special public funds" as the theme search terms, and searches on the database of "PKULAW", the official websites of provincial governments, and the official websites of provincial departments of education and finance. This study collected a total of 13 provincial government departments issued, the current effective private colleges and universities provincial special public funding policy texts. The basic information of the policy samples is shown in Table 1. It can be found that these policy texts are basically the management methods of provincial governments for the provincial special funds of private colleges and universities, and these 13 provinces basically cover the eastern, central, western and northeastern regions of China.

**Table 1. Sample policies for PMC-Index model.**

| Item | Policy name | Release time | Release agency |
|---|---|---|---|
| P1 | Measures of Shanghai Municipal Special Fund for Promoting the Development of Private Education | 2016. 03 | Shanghai Municipal Education Commission and Finance Bureau |
| P2 | Measures for the Administration of Special Funds for the Development of Private Higher Education in Fujian Province | 2017. 10 | Fujian Provincial Department of Finance and Department of Education |
| P3 | Measures for the Administration of special funds for the development of provincial private education in Sichuan Province | 2014. 11 | Sichuan Provincial Department of Finance and Department of Education |
| P4 | Measures for Management of Financial Support Funds for Colleges and Universities run by Chongqing Citizens | 2016. 06 | Chongqing Education Commission and Finance bureau |
| P5 | Measures for the Management of Special Subsidies for Private Education in Inner Mongolia Autonomous Region | 2015. 11 | Inner Mongolia Autonomous Region Department of Finance and Department of Education |
| P6 | Measures for the Administration of Special Funds for the Development of Private Education in Ningxia Hui Autonomous Region | 2019. 01 | Ningxia Hui Autonomous Region Department of Finance and Department of Education |
| P7 | Interim Measures of Jiangsu Provincial Special Fund Management for the Development of Private Higher Education | 2018. 06 | Jiangsu Provincial Department of Finance and Department of Education |
| P8 | Measures for the Administration of Development Support Funds of Provincial Private Colleges and Universities of Jilin Province | 2020. 07 | Jilin Provincial Department of Finance and Department of Education |
| P9 | Measures for the Administration of Provincial Special Funds for the Development of Private Education in Guizhou Province | 2021. 10 | Guizhou Provincial Department of Finance and Department of Education |
| P10 | Measures for Fund Management of Basic Capacity Construction of Private Colleges and Universities of Shandong Province | 2018. 08 | Shandong Provincial Department of Education and Department of Finance |
| P11 | Interim Measures for the Administration of Special funds for the development of Private Education in Jiangxi Province | 2015. 09 | Jiangxi Provincial Department of Finance and Department of Education |
| P12 | Interim Measures of Henan Province on the Administration of the Special Fund for the Development of Private Education | 2013. 05 | Henan Provincial Department of Education |
| P13 | Administrative Measures of the Special Fund for the Development of Private Education in Hunan Province | 2011. 03 | Hunan Provincial Department of Education |

## Methods of analysis

In this study, the text mining method and PMC index model are used to quantitatively evaluate the content of the provincial special public funding policy of private universities in China. Among them, text mining is a kind of information analysis technology that finds and extracts valuable knowledge from unstructured texts through patterns and relationships, aiming at revealing facts, trends or structures [39]. In this paper, the correlation analysis function is mainly applied. In the "Classification of variables and identification of parameters" stage of PMC-Index model, the word frequency of 13 policy text keywords is counted by text mining tools, and the semantic network diagram of high-frequency keywords is displayed in a visual form, so as to provide a basis for the determination of primary and secondary evaluation indicators. PMC-Index model is a quantitative evaluation method for a single policy based on the content of the policy text proposed by Ruiz Estrada and other scholars based on the Omnia Mobilis hypothesis [40]. The Omnia Mobilis hypothesis emphasizes that everything in the world is constantly moving and connected, and the evaluation of a thing should be comprehensive and should not ignore any possible relevant variables [41]. Therefore, PMC index model differs from other policy evaluation models in that it uses binary 0 and 1 to balance all variables, and emphasizes that the number and weight of variables should not be limited. This model has two characteristics. First, it can analyze the internal consistency of policies from multiple dimensions. Second, it can visualize the advantages and disadvantages of the strategy text by calculating the PMC-Index and constructing the PMC-Surface.

## Construction of the PMC-Index model

In general, the construction of PMC-Index model consists of the following four steps (Fig 1).

**Classification of variables and identification of parameters.** On the basis of adequately reading and understanding the original policy text, this paper extracts the high-frequency keywords of 13 policy texts with the help of commonly used text analysis tools, and constructs the semantic network diagram of high-frequency keywords, so as to provide a basis for the determination of primary and secondary evaluation indicators. First of all, this study uses text mining software ROSTCM6 (Wuhan University, Hubei, China) to extract high-frequency keywords. In this study, 13 policy texts were imported into the text mining database of ROSTCM6 software to form document sets for text segmentation processing, and word frequency statistics were performed on the document sets after word segmentation. The output word segmentation results were displayed in order of word frequency from high to low. In order to ensure the ideal statistical results, the word segmentation list needs to be redefined and the proper terms such as "private universities" and "special funds" should be added in the word segmentation processing. In addition, since the policy samples are basically the management methods of the special funds for the development of private education, the occurrence frequency of words such as "private education" and "special funds" is relatively high. In the analysis of policy characteristics, such words are superfluous and have no obvious effect on the results. Moreover, high-frequency policy words such as "requirements" and "acceleration", which frequently appear, have no practical significance for policy analysis. So they need to be filtered out. After the above words are eliminated, the top 60 effective high-frequency keywords with the highest frequency are selected for analysis and reference (Table 2).

Secondly, this study uses the complex network analysis software Gephi to build the semantic network diagram of high-frequency keywords (Fig 2). Nodes in the figure represent high-frequency keywords, and the thickness of the connection between nodes reflects the strength of the relationship between them. The thicker the line, the stronger the correlation between the keywords. In the network diagram, the keywords of the text of the provincial special public funding policy of China's private colleges and universities are combined in the form of a network, which can directly reflect the structural relationship between the keywords and provide an important reference for variable setting. As can be seen from the figure, "finance", "funds", "use" and "management" are located in the center of the network and have a high connection density with other keywords, which indicates that the use and management of financial funds is the core theme of this kind of policy text. Around this theme, the policy text clarifies the fund management "departments" such as "private schools", "Department of Education" and "Department of Finance", and emphasizes the key links of fund management such as "budget arrangement", "use of funds", "project declaration", "allocation" of funds, "performance evaluation" and "supervision". In addition, the keywords such as "promote", "standardize", "develop" and "run a school" located at the edge of the network indicate that the long-term goal of such policies is to promote the standardized running of private colleges and universities and healthy development.

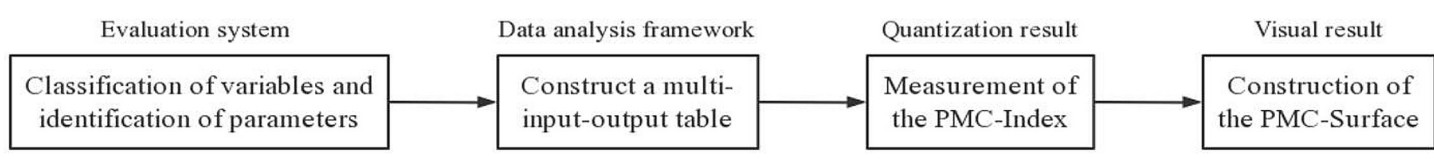

**Fig 1. Construction framework of the PMC-Index model.**

**Table 2. Statistical frequency of effective words from the policy texts.**

| Vocabulary | Frequency | Vocabulary | Frequency | Vocabulary | Frequency |
|---|---|---|---|---|---|
| funds | 282 | declaration | 64 | implementation | 35 |
| project | 205 | measures | 58 | goals | 32 |
| administration | 194 | expenditure | 54 | financial | 31 |
| school | 189 | evaluation | 53 | system | 30 |
| development | 155 | promotion | 52 | teaching | 30 |
| education | 140 | according to | 52 | disburse | 29 |
| private | 138 | provincial | 51 | review | 28 |
| finance | 142 | arrangement | 46 | input | 27 |
| school running | 129 | supervision | 46 | in accordance with the law | 26 |
| use | 129 | specification | 46 | societies | 25 |
| department of education | 112 | factors | 45 | key | 25 |
| performance | 110 | awards | 45 | materials | 24 |
| support | 99 | conditions | 44 | standard | 24 |
| construction | 90 | subsidies | 43 | reform | 24 |
| assignment | 89 | acts | 43 | set | 23 |
| budget | 83 | application | 43 | carry out | 23 |
| department | 81 | examine | 39 | programs | 23 |
| department of finance | 77 | organization | 39 | plan | 23 |
| regulations | 76 | inspection | 39 | finance bureau | 22 |
| annual | 73 | transmit to lower levels | 37 | formulation | 22 |

On the basis of the above analysis, this paper combined with the characteristics of the provincial special public funding policy of China's private colleges and universities, referred to the scholars' relevant research on the management of special funds of private colleges and universities [42–45], and established an evaluation index system of the provincial special public funding policy of China's private colleges and universities under the guidance of experts. A total of 9 primary variables and 40 Secondary variables are set in this paper, which fully covers the content and information of the special public funding policy document.

After the selection and classification of the variables, the parameters of each variable need to be set. To ensure that all variables have the same importance and weight level. In this paper, the evaluation parameters of secondary variables under $X_1$ (Policy type), $X_3$ (Responsibilities regulations), $X_4$ (Sustainability of funds), $X_5$ (Use of funds), $X_6$ (Scope of assistance), $X_7$ (Conditions of application), $X_8$ (Allocation of funds) and $X_9$ (capital regulation) are set to obey the binomial distribution of [0, 1]. Specifically, if there is any content in the text of the provincial special public funding policy of China's private colleges and universities that meets the evaluation criteria of the second-level variable, the parameter of the second-level variable is "1"; otherwise, it is "0". It should be noted that this approach does not apply to variables whose evaluation criteria are mutually exclusive [46]. In order to overcome this defect of the traditional PMC-Index model, the evaluation parameters of the secondary variable $X_2:3$ (Enforcement effectiveness) under the Primary variable $X_2$ (Policy effectiveness) are set as [0. 5, 1] (Table 3).

**Construct a multi-input-output table.** In order to better quantify the value of each subvariable, the second step of the PMC-Index model needs to establish a multi-input output table (Table 4). Multi-input-output table is a data analysis framework that can store a large amount of data and measure single variables in multiple dimensions [47]. Constructing multi-

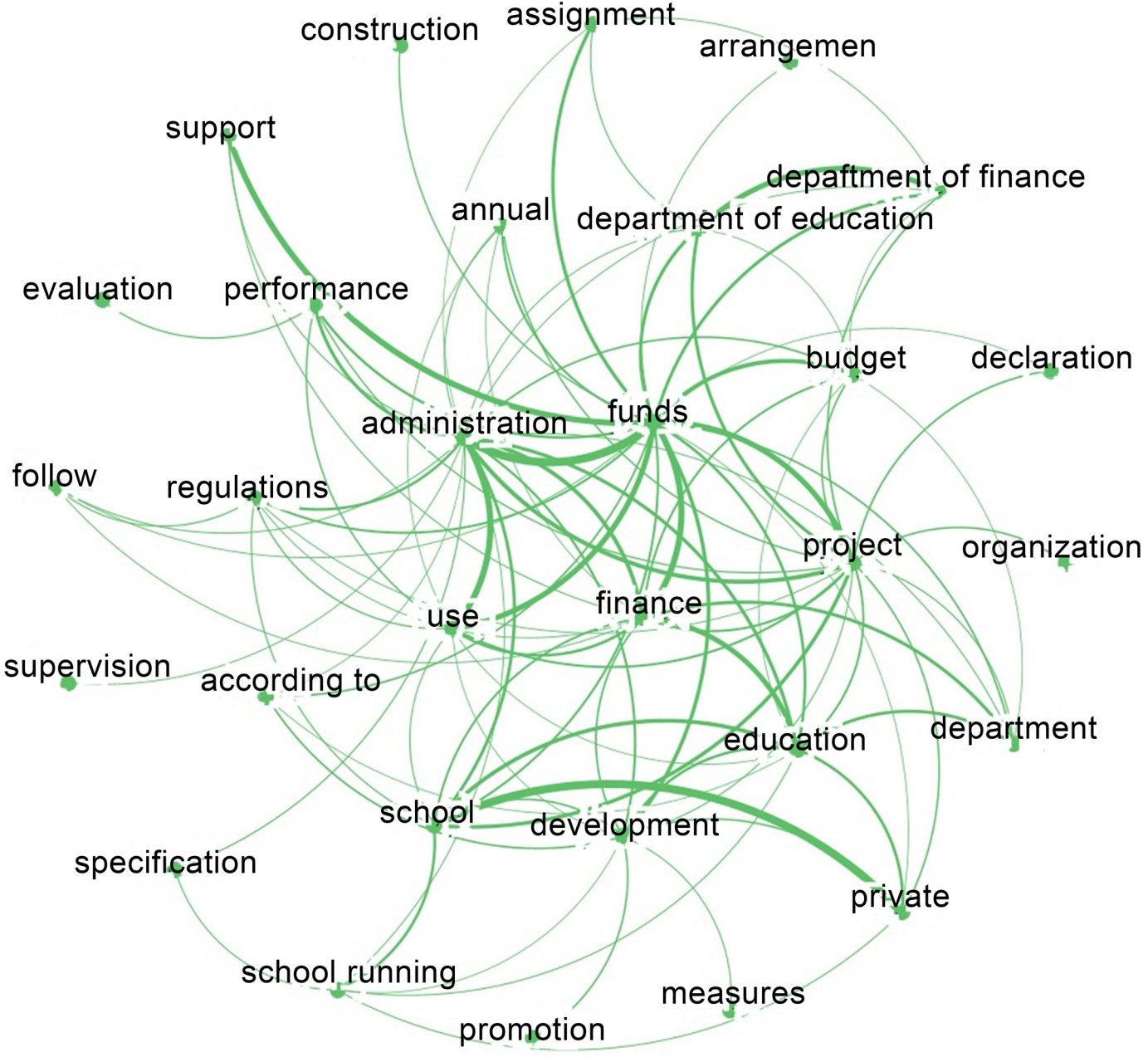

**Fig 2. Network graph of the high-frequency theme words.**

input-output table is the basis of calculating variable index weight, and it is a necessary step in the process of establishing PMC-Index model.

**Measurement of the PMC-Index.** The PMC-Index is usually calculated by the following four steps. Firstly, nine first-level variables and 40 second-level variables in the provincial special public funding policy of private colleges and universities are put into the multi-input-output Table 4. Secondly, text mining and formulas (1) and (2) are used to assign values to multiple secondary variables under the same primary variable. Thirdly, formula (3) is used to calculate the values of each first-level variable. Fourthly, formula (4) is used to calculate the

**Table 3. Variable selection and evaluation criteria of PMC-Index model evaluation.**

| Primary variable | Secondary variable | Evaluation criteria for secondary variables | Evaluation parameter |
|---|---|---|---|
| Policy type ($X_1$) | Guidance($X_1$:1) | Whether the policy is instructive | $X_1$:3~N[0,1] |
| | Supervision($X_1$:2) | whether the policy involves regulation | $X_1$:4~N[0,1] |
| | Others($X_1$:3) | Whether the policy has other meaning | $X_1$:5~N[0,1] |
| Policy effectiveness ($X_2$) | Policy objective($X_2$:1) | Whether the policy has a clear policy objective | $X_2$:1~N[0,1] |
| | Timeliness of policies($X_2$:2) | Whether the policy has a clear deadline | $X_2$:2~N[0,1] |
| | Enforcement effectiveness($X_2$:3) | Whether the policy is formal (temporary) | $X_2$:3~N[0. 5,1] |
| Responsibilities regulations ($X_3$) | Responsibilities of education department ($X_3$:1) | Whether the policy clarifies the responsibilities of the education sector | $X_3$:1~N[0,1] |
| | Responsibility of financial department ($X_3$:2) | Whether the policy clarifies the responsibilities of the financial sector | $X_3$:2~N[0,1] |
| | Responsibilities of private colleges and universities($X_3$:3) | Whether the policy has clarified the responsibilities of private colleges and universities | $X_3$:3~N[0,1] |
| Sustainability of funds ($X_4$) | Legal basis($X_4$:1) | Whether the policy specifies the legal basis for special funds | $X_4$:1~N[0,1] |
| | Source of funds($X_4$:2) | Whether the policy clearly defines the funding source of the special funds | $X_4$:2~N[0,1] |
| | budget layout($X_4$:3) | Whether the policy specifies the budget arrangement of special funds | $X_4$:3~N[0,1] |
| Use of funds ($X_5$) | Principles of use($X_5$:1) | Whether the policy specifies the principles for the use of special funds | $X_5$:1~N[0,1] |
| | management style($X_5$:2) | Whether the policy is clear about the management style of the special funds | $X_5$:2~N[0,1] |
| | Applicable object($X_5$:3) | Whether the policy specifies the object of application of special funds | $X_5$:3~N[0,1] |
| Scope of assistance ($X_6$) | Party building and ideological and political work($X_6$:1) | Whether the scope of assistance required by the policy includes party building and ideological and political work | $X_6$:1~N[0,1] |
| | Conditions for running schools($X_6$:2) | Whether the scope of assistance required by the policy includes school conditions | $X_6$:2~N[0,1] |
| | Discipline construction($X_6$:3) | Whether the scope of assistance required by the policy includes the discipline construction | $X_6$:3~N[0,1] |
| | Teaching staff($X_6$:4) | Whether the scope of assistance required by the policy includes the teaching staff | $X_6$:4~N[0,1] |
| | Reform project($X_6$:5) | Whether the scope of assistance required by the policy includes reform projects | $X_6$:5~N[0,1] |
| | Public service($X_6$:6) | Whether the scope of assistance required by the policy includes public services | $X_6$:6~N[0,1] |
| | Recognition and reward($X_6$:7) | Whether the scope of assistance required by the policy includes recognition and awards | $X_6$:7~N[0,1] |
| | Prohibited items($X_6$:8) | Whether the scope of assistance required by the policy includes prohibited items | $X_6$:8~N[0,1] |
| Conditions of application ($X_7$) | Insisting public welfare under running school($X_7$:1) | Whether the application conditions stipulated by the policy involve Insisting public welfare under running school | $X_7$:1~N[0,1] |
| | Conduct of running schools should be standardized($X_7$:2) | Whether the application conditions stipulated by the policy involve standardized school-running behavior | $X_7$:2~N[0,1] |
| | Conditions for running schools should meet standards($X_7$:3) | Whether the application conditions stipulated by the policy involve the requirements of school running conditions | $X_7$:3~N[0,1] |
| | The scale of schools should meet the standards($X_7$:4) | Whether the application conditions stipulated by the policy involve the requirements of the size of the school | $X_7$:4~N[0,1] |
| | Staff hiring should be standardized($X_7$:5) | Whether the application conditions specified in the policy involve the standardized employment of faculty members | $X_7$:5~N[0,1] |
| | Financial management should be standardized($X_7$:6) | Whether the application conditions specified in the policy involve standardized financial management | $X_7$:6~N[0,1] |
| | The property rights of legal persons should be implemented($X_7$:7) | Whether the application conditions specified in the policy involve the implementation of the property rights of legal persons | $X_7$:7~N[0,1] |
| | Prohibition situation($X_7$:8) | Whether the application conditions specified in the policy involve prohibited circumstances | $X_7$:8~N[0,1] |

(*Continued*)

**Table 3.** (Continued)

| Primary variable | Secondary variable | Evaluation criteria for secondary variables | Evaluation parameter |
|---|---|---|---|
| Allocation of funds ($X_8$) | Allocation method($X_8$:1) | Whether the policy specifies the allocation method of special funds | $X_8$:1~N[0,1] |
| | Review procedure($X_8$:2) | Whether the policy specifies the review procedures for special funds | $X_8$:2~N[0,1] |
| | fund appropriation($X_8$:3) | Whether the policy specifies the appropriations procedures for special funds | $X_8$:3~N[0,1] |
| Fund regulation ($X_9$) | supervising subject($X_9$:1) | Whether the policy clearly defines the subject of special fund supervision | $X_9$:1~N[0,1] |
| | funds are used for specified purposes only ($X_9$:2) | Whether the policy makes it clear that special funds are used for specified purposes only | $X_9$:2~N[0,1] |
| | Execution time frame ($X_9$:3) | Whether the policy specifies the time limit for the implementation of special funds | $X_9$:3~N[0,1] |
| | Assets ownership ($X_9$:4) | Whether the policy specifies the ownership of assets acquired through the use of special funds | $X_9$:4~N[0,1] |
| | Violation processing($X_9$:5) | Whether the policy specifies how to deal with the improper use of special funds | $X_9$:5~N[0,1] |
| | Performance evaluation($X_9$:6) | Whether the policy specifies the performance evaluation of special funds | $X_9$:6~N[0,1] |

PMC-Index of each policy to be evaluated. With reference to the existing research, the policies are classified into different levels according to the PMC-Index score. The classification standards are shown in Table 5.

$$X \sim N[0, 1] \tag{1}$$

$$X = \{XR : [0 \sim 1]\} \tag{2}$$

$$X_t = \sum_{j=1}^{n} \frac{x_{tj}}{T\left(x_{tj}\right)} \tag{3}$$

$$PMC - Index = \sum_{t=1}^{m} X_t \tag{4}$$

Where, t is the main-variable, i = 1, 2, 3, $\cdots$, m. j is the sub-variable, j = 1, 2, $\cdots$, n. T is the number of the total sub-variables in analysis.

**Construction of the PMC-Surface.** PMC-Surface is a visual processing of PMC-Index, which can more intuitively display the pros and cons of policies in various dimensions. PMC-Matrix is the basis of PMC-Surface rendering, and its construction is shown in Formula (5).

**Table 4. Multi-input-output table.**

| | |
|---|---|
| $X_1$ | $X_1$:1 $X_1$:2 $X_1$:3 |
| $X_2$ | $X_2$:1 $X_2$:2 $X_2$:3 |
| $X_3$ | $X_3$:1 $X_3$:2 $X_3$:3 |
| $X_4$ | $X_4$:1 $X_4$:2 $X_4$:3 |
| $X_5$ | $X_5$:1 $X_5$:2 $X_5$:3 |
| $X_6$ | $X_6$:1 $X_6$:2 $X_6$:3 $X_6$:4 $X_6$:5 $X_6$:6 $X_6$:7 $X_6$:8 |
| $X_7$ | $X_7$:1 $X_7$:2 $X_7$:3 $X_7$:4 $X_7$:5 $X_7$:6 $X_7$:7 $X_7$:8 |
| $X_8$ | $X_8$:1 $X_8$:2 $X_8$:3 |
| $X_9$ | $X_9$:1 $X_9$:2 $X_9$:3 $X_9$:4 $X_9$:5 $X_9$:6 |

Table 5. Policy level classification criteria based on PMC-Index.

| PMC-Index | 0–4. 99 | 5–6. 99 | 7–7. 99 | 8–9 |
|---|---|---|---|---|
| Evaluation level | Unacceptable(U) | Acceptable(A) | Excellent(E) | Perfect(P) |

$$PMC - Surface = \begin{bmatrix} X_1 & X_2 & X_3 \\ X_4 & X_5 & X_6 \\ X_7 & X_8 & X_9 \end{bmatrix} \quad (5)$$

## Results and analysis

### Empirical results

Based on the PMC-Index evaluation model constructed above, this paper obtained the multi-input output table of 13 policy samples(Table 6) by text mining technology. On this basis, this paper calculated the PMC-Index (two decimal places are retained) and its ranking and grade of each policy (Table 7), and further drew the PMC-Surface of some policies (Fig 3). Due to the limited space, this paper only selected the typical policies with the highest, lowest PMC-Index scores, the middle level of acceptable grade and the middle level of excellent grade among 13 policy samples as examples to show their PMC-Surface graph. The smoother the surface diagram is, the higher the consistency level of the policy is, and the development of the policy in different aspects is more balanced. If the convex and convex amplitude of the surface map is larger, the consistency level of the policy is worse, and the level of development intensity of the policy is different in different aspects [48].

### Holistic analysis

From the overall performance of various policies in Table 7, the average PMC-Index of 13 policies is 6.97, indicating that the policy consistency is at an acceptable level. The 13 policies based on the PMC-Index were ranked as P1> P9> P8> P6> P5> P2> P10> P3> P4> P11> P12> P7> P13. According to the grading criteria in Table 5, among the 13 policies, there are 2 perfect grade policies, 5 excellent grade policies, 5 acceptable grade policies, and 1 unacceptable grade policies. Thus, the overall quality of the 13 policy is relatively good, and the policy content have a certain completeness and rationality. However, there are still unacceptable grade policies, which indicates that the content design of sample policies still has great room for improvement.

From the mean of the scores on the various Primary variables, the average scores of $X_1$ (policy type), $X_2$ (policy effectiveness), $X_4$ (Sustainability of funds), $X_5$ (Use of funds), $X_8$ (Allocation of funds) and $X_9$ (Fund regulation) are relatively high, indicating that local government departments have paid more attention to these variables when formulating relevant policies, and considered all dimensions of these variables more comprehensively. However, the mean values of Primary variables $X_3$ (Responsibilities regulations), $X_6$ (Scope of assistance) and $X_7$ (Conditions of application) are low, indicating that the content design of these policies still has obvious defects in these three aspects and needs to make up for the shortcomings. In addition, from the morphology of the average PMC-Surface graph in Fig 3, it can be seen that the PMC-Surface of the 13 policies is relatively smooth as a whole, which indicates that the average

**Table 6. Multi-input-output table of 13 policy samples.**

| | | P_1 | P_2 | P_3 | P_4 | P_5 | P_6 | P_7 | P_8 | P_9 | P_10 | P_11 | P_12 | P_13 |
|---|---|---|---|---|---|---|---|---|---|---|---|---|---|---|
| $X_1$ | $X_1$:1 | 1 | 1 | 1 | 1 | 1 | 1 | 1 | 1 | 1 | 1 | 1 | 1 | 1 |
| | $X_1$:2 | 1 | 1 | 1 | 1 | 1 | 1 | 1 | 1 | 1 | 1 | 1 | 1 | 1 |
| | $X_1$:3 | 1 | 1 | 1 | 1 | 1 | 1 | 1 | 1 | 1 | 1 | 1 | 1 | 1 |
| $X_2$ | $X_2$:1 | 1 | 1 | 1 | 1 | 1 | 1 | 1 | 1 | 1 | 1 | 1 | 1 | 1 |
| | $X_2$:2 | 0 | 0 | 1 | 0 | 1 | 1 | 0 | 1 | 1 | 1 | 0 | 0 | 0 |
| | $X_2$:3 | 1 | 1 | 1 | 1 | 1 | 1 | 0.5 | 1 | 1 | 1 | 1 | 0.5 | 0.5 |
| $X_3$ | $X_3$:1 | 1 | 1 | 0 | 0 | 1 | 1 | 1 | 1 | 1 | 0 | 1 | 0 | 0 |
| | $X_3$:2 | 1 | 1 | 0 | 0 | 1 | 1 | 1 | 1 | 1 | 0 | 1 | 0 | 0 |
| | $X_3$:3 | 1 | 1 | 0 | 0 | 0 | 0 | 0 | 1 | 0 | 1 | 0 | 0 | 0 |
| $X_4$ | $X_4$:1 | 1 | 1 | 1 | 1 | 1 | 1 | 1 | 1 | 1 | 1 | 1 | 1 | 0 |
| | $X_4$:2 | 1 | 1 | 1 | 1 | 1 | 1 | 1 | 1 | 1 | 1 | 1 | 1 | 0 |
| | $X_4$:3 | 1 | 1 | 1 | 1 | 1 | 1 | 1 | 1 | 1 | 1 | 0 | 1 | 0 |
| $X_5$ | $X_5$:1 | 1 | 0 | 1 | 1 | 1 | 1 | 1 | 1 | 1 | 1 | 0 | 1 | 1 |
| | $X_5$:2 | 1 | 1 | 1 | 1 | 1 | 1 | 1 | 1 | 1 | 1 | 1 | 1 | 0 |
| | $X_5$:3 | 1 | 1 | 1 | 1 | 1 | 1 | 1 | 1 | 1 | 1 | 1 | 1 | 1 |
| $X_6$ | $X_6$:1 | 0 | 0 | 0 | 0 | 0 | 0 | 0 | 0 | 0 | 0 | 0 | 0 | 0 |
| | $X_6$:2 | 1 | 1 | 1 | 1 | 1 | 1 | 1 | 1 | 1 | 1 | 1 | 0 | 1 |
| | $X_6$:3 | 1 | 1 | 1 | 1 | 1 | 1 | 0 | 1 | 1 | 1 | 1 | 1 | 0 |
| | $X_6$:4 | 1 | 1 | 0 | 1 | 1 | 1 | 0 | 1 | 1 | 1 | 1 | 1 | 0 |
| | $X_6$:5 | 1 | 0 | 0 | 1 | 0 | 0 | 0 | 1 | 1 | 0 | 0 | 1 | 0 |
| | $X_6$:6 | 1 | 0 | 1 | 0 | 0 | 0 | 1 | 0 | 0 | 1 | 0 | 0 | 0 |
| | $X_6$:7 | 1 | 1 | 1 | 0 | 0 | 1 | 1 | 1 | 1 | 0 | 0 | 1 | 1 |
| | $X_6$:8 | 1 | 1 | 0 | 1 | 0 | 1 | 1 | 1 | 1 | 1 | 1 | 1 | 0 |
| $X_7$ | $X_7$:1 | 1 | 0 | 0 | 0 | 0 | 0 | 0 | 0 | 1 | 0 | 0 | 0 | 0 |
| | $X_7$:2 | 1 | 1 | 1 | 0 | 1 | 1 | 0 | 1 | 1 | 1 | 1 | 1 | 1 |
| | $X_7$:3 | 1 | 0 | 0 | 0 | 1 | 1 | 0 | 0 | 1 | 1 | 1 | 1 | 1 |
| | $X_7$:4 | 0 | 0 | 0 | 0 | 1 | 0 | 0 | 0 | 1 | 0 | 1 | 1 | 0 |
| | $X_7$:5 | 1 | 1 | 1 | 0 | 0 | 0 | 0 | 0 | 1 | 1 | 0 | 0 | 1 |
| | $X_7$:6 | 1 | 1 | 1 | 0 | 0 | 1 | 0 | 1 | 1 | 1 | 0 | 0 | 1 |
| | $X_7$:7 | 1 | 1 | 1 | 0 | 1 | 1 | 0 | 0 | 1 | 1 | 0 | 0 | 1 |
| | $X_7$:8 | 1 | 0 | 0 | 1 | 0 | 1 | 0 | 1 | 0 | 0 | 0 | 0 | 1 |
| $X_8$ | $X_8$:1 | 1 | 1 | 1 | 1 | 1 | 1 | 1 | 1 | 1 | 1 | 1 | 1 | 1 |
| | $X_8$:2 | 1 | 1 | 1 | 1 | 1 | 1 | 0 | 1 | 1 | 1 | 1 | 1 | 1 |
| | $X_8$:3 | 1 | 1 | 1 | 1 | 1 | 1 | 0 | 1 | 1 | 0 | 1 | 1 | 0 |
| $X_9$ | $X_9$:1 | 1 | 1 | 1 | 1 | 1 | 1 | 1 | 1 | 1 | 1 | 1 | 1 | 0 |
| | $X_9$:2 | 1 | 1 | 1 | 1 | 1 | 1 | 1 | 1 | 1 | 1 | 1 | 1 | 1 |
| | $X_9$:3 | 1 | 0 | 0 | 1 | 1 | 0 | 0 | 1 | 1 | 1 | 0 | 0 | 0 |
| | $X_9$:4 | 1 | 1 | 1 | 1 | 1 | 1 | 0 | 0 | 1 | 1 | 1 | 1 | 0 |
| | $X_9$:5 | 1 | 1 | 1 | 1 | 1 | 1 | 1 | 1 | 1 | 1 | 1 | 1 | 1 |
| | $X_9$:6 | 1 | 1 | 1 | 1 | 1 | 1 | 1 | 1 | 1 | 1 | 1 | 1 | 0 |

level of internal consistency of the 13 policy samples is high, and the development intensity of the policies in various aspects is balanced and the structure is reasonable.

From the perspective of the time distribution of the introduction of policies, the introduction of the 13 provincial special public funding policies for China's private colleges and universities has a large span of time, and there is no phenomenon of centralized release. It can be found that the PMC-Index scores of these policies show a positive correlation trend with the

**Table 7. PMC-Index and level of 13 policies.**

|  | P1 | P2 | P3 | P4 | P5 | P6 | P7 | P8 | P9 | P10 | P11 | P12 | P13 | Mean value |
|---|---|---|---|---|---|---|---|---|---|---|---|---|---|---|
| $X_1$ | 1 | 1 | 1 | 1 | 1 | 1 | 1 | 1 | 1 | 1 | 1 | 1 | 1 | 1 |
| $X_2$ | 0.67 | 0.67 | 1 | 0.67 | 1 | 1 | 0.5 | 1 | 1 | 1 | 0.67 | 0.5 | 0.67 | 0.80 |
| $X_3$ | 1 | 1 | 0 | 0 | 0.67 | 0.67 | 0.67 | 1 | 0.67 | 0.33 | 0.67 | 0 | 0 | 0.51 |
| $X_4$ | 1 | 1 | 1 | 1 | 1 | 1 | 1 | 1 | 1 | 1 | 0.67 | 1 | 0 | 0.90 |
| $X_5$ | 1 | 0.67 | 1 | 1 | 1 | 1 | 1 | 1 | 1 | 1 | 0.67 | 1 | 0.67 | 0.92 |
| $X_6$ | 0.88 | 0.63 | 0.5 | 0.63 | 0.38 | 0.63 | 0.5 | 0.75 | 0.75 | 0.63 | 0.5 | 0.63 | 0.25 | 0.59 |
| $X_7$ | 0.88 | 0.5 | 0.5 | 0.13 | 0.5 | 0.63 | 0 | 0.38 | 0.88 | 0.63 | 0.38 | 0.38 | 0.75 | 0.50 |
| $X_8$ | 1 | 1 | 1 | 1 | 1 | 1 | 0.33 | 1 | 1 | 0.67 | 1 | 1 | 0.67 | 0.90 |
| $X_9$ | 1 | 0.83 | 0.83 | 1 | 1 | 0.83 | 0.67 | 0.83 | 1 | 1 | 0.83 | 0.83 | 0.33 | 0.84 |
| PMC-Index | 8.43 | 7.30 | 6.83 | 6.43 | 7.55 | 7.76 | 5.67 | 7.96 | 8.30 | 7.26 | 6.39 | 6.34 | 4.34 | 6.96 |
| Rank | 1 | 6 | 8 | 9 | 5 | 4 | 12 | 3 | 2 | 7 | 10 | 11 | 13 | |
| Level | P | E | A | A | E | E | A | E | P | E | A | A | U | |

publication time. This is in line with the development law of China's private higher education policy contents becoming more and more scientific and perfect. Particularly, after 2016, the special public funding policies issued by various localities scored significantly higher, because the second amendment of *the Law of the People's Republic of China on the Promotion of Private Education* in 2016 provided a legal basis for the financial differential support of the two types of private colleges and universities, making the public funding policies of private education more institutionalized and intensified. In addition, P13, P12 and other special funding policies have not been revised or improved for nearly ten years, and the PMC-Index score is significantly lower. This kind of policy needs to be improved in time to improve the completeness and rationality of the policy content in various variables and dimensions.

## Various policy analysis

In order to further clarify the characteristics and problems of the content design of the provincial special public funding policies of China's private colleges and universities, this section makes a detailed analysis of the evaluation of 13 policy samples with the help of the multi-input-output table (Table 6).

**1. Perfect level policy.** P1 and P9 were rated as perfect. Both of them are policy documents issued or revised after 2016, and the scores of Primary variables $X_1$ (policy type), $X_4$ (Sustainability of funds), $X_5$ (Use of funds), $X_8$ (Allocation of funds) and $X_9$ (Fund regulation) all reach the highest value, and almost all Primary variables score above the mean value. This shows that the content design of the perfect level policy is relatively scientific and reasonable, and the policy dimension is more comprehensive and consistent. However, P1 and P9 still have the phenomenon of losing points, leaving some room for improvement. P1 lacks a clear deadline in $X_2$ (Policy effectiveness), which is not conducive to the continuous revision and improvement of the policy, and the timeliness of the policy is insufficient; In $X_6$ (Scope of assistance), the relevant content of party building is not involved; In $X_7$ (Conditions of application), it is also necessary to strengthen the limited requirements on the scale of private colleges and universities. P9 does not clearly specify the corresponding responsibilities of private colleges and universities in $X_3$ (Responsibilities regulations); $X_6$ (Scope of assistance) does not involve the content of party building and public service; In $X_7$ (Conditions of application) is not introduced prohibited application situation. It can be found that the two policies have both individual differences and common deficiencies, and the main problems focus on $X_6$ and $X_7$.

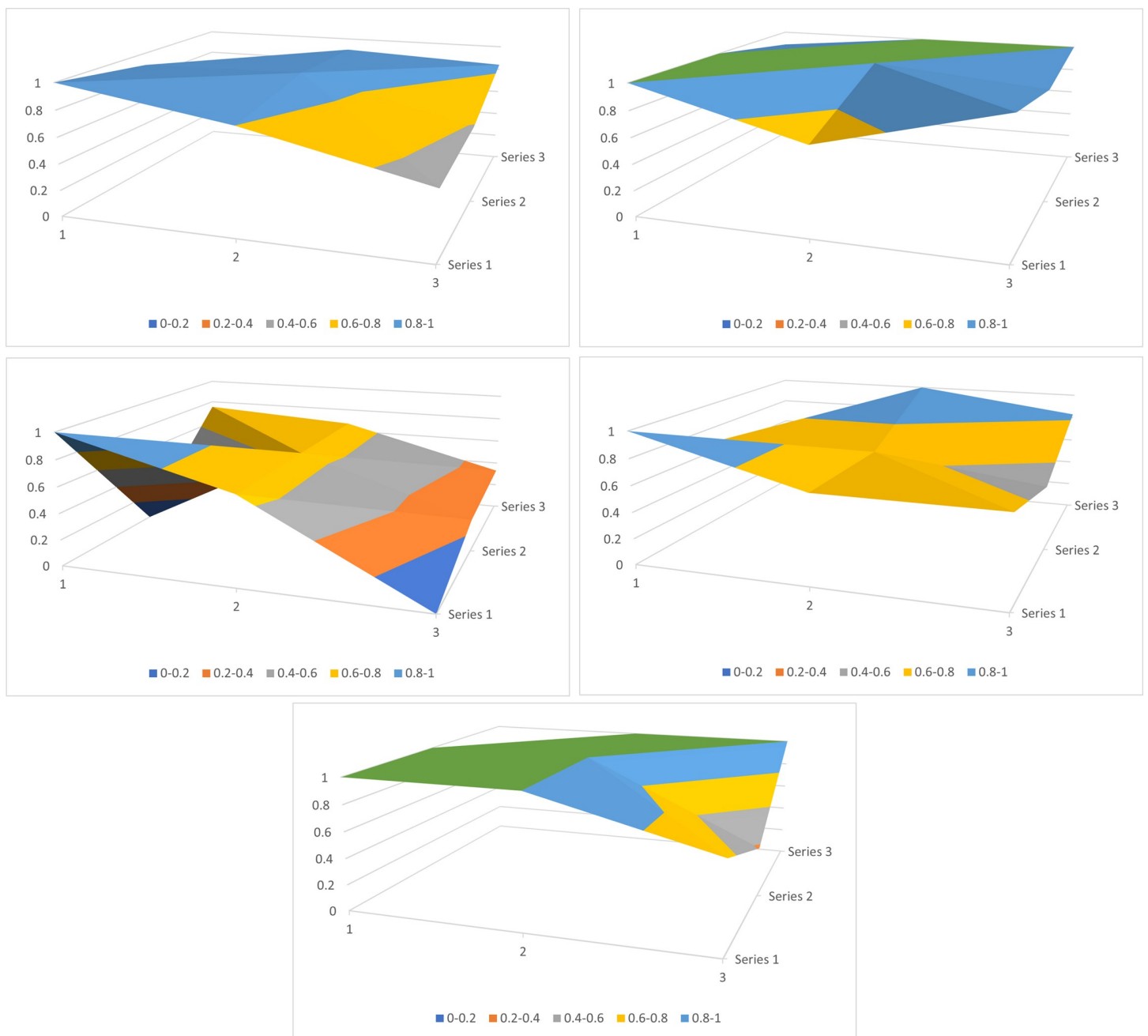

**Fig 3. PMC-Surface of representative policy.** (a). Average PMC-Surface for all policy samples. (b). The PMC-Surface of P1. (c). The PMC-Surface of P13. (d).The PMC-Surface of P11. (e).The PMC-Surface of P5.

**2. Excellent level policy.** P2, P5, P6, P8 and P10 are rated as excellent. In general, the PMC-Index scores of excellent policies only reached the highest value in $X_1$ (Policy type) and $X_4$ (Sustainability of funds), and had relatively high scores in $X_2$ (Policy effectiveness), $X_5$ (Use of funds), $X_8$ (Allocation of funds) and $X_9$ (Fund regulation), with an average score above 0.89. However, the average score of excellent policies in $X_3$ (Responsibilities regulations), $X_6$ (Scope of assistance), and $X_7$ (Conditions of application) is relatively low, which needs breakthrough.

Specifically, in $X_2$ and $X_5$, P2 does not make a clear deadline for the policy, nor does it establish the principle of the use of special funds, ignoring the timeliness of the policy and the direction of the use of funds. In $X_8$, P10 does not clearly specify the appropriations procedures for special funds, which will easily cause the situation that the special funds are difficult to be in place or slow to be allocated. In $X_9$, P8 ignores the ownership of fixed assets obtained by private colleges and universities using special funds, which is not conducive to the implementation of legal person property rights in private colleges and universities. P2 and P6 do not make corresponding provisions on the implementation time limit of special funds, which is difficult to improve the efficiency of the use of funds at the implementation level.

In $X_3$, P5 and P6 are similar to the perfect level policy P9, which does not specify the relevant responsibilities of private colleges and universities, while P10 does not specifically point out the relevant responsibilities of provincial education departments and financial departments. The education department, financial department and private colleges and universities are the important supervisory bodies of the special funds, and the ambiguity of their management responsibilities will inevitably bring about the dislocation and disconnection in the application, use and distribution of funds. In $X_6$, the average score of the PMC-Index of the five policies is 0.604, indicating that the support scope stipulated by excellent policies is relatively narrow. Among them, P5 scored only 0.38 on this variable, which is far below the average. P5 only focuses on school-running conditions, discipline construction and teaching staff, and its vision is too focused and narrow, and it does not specify projects that are prohibited from using special funds. In $X_7$, the average PMC-Index score of excellent level policies is only 0.528, which is the lowest among the nine primary variables and requires high attention from policy makers. In the content design of this variable, P8 does not stipulate the public welfare of private colleges and universities, the conditions of running a school, the scale of running a school, the employment norms of faculty and staff, and the implementation of the property rights of legal persons, which makes it difficult to give full play to the guiding and regulating role of special funds in the healthy and sustainable development of private colleges and universities.

**3. Acceptable level policy.** P3, P4, P7, P11 and P12 were rated as acceptable. The PMC-Index scores for these policies are all lower than the overall average. At the same time, except for $X_1$, $X_4$ and $X_5$, the average score of PMC-Index on other primary variables is also lower than the overall average. This shows that the overall quality of acceptable level policies is general and the consistency is poor, and policy makers need to comprehensively review, adjust and improve the content of policies.

Specifically, there are many common problems between acceptable policies, perfect policies and excellent policies, such as the incomplete setting of $X_6$ and $X_7$; The absence of a time limit for implementation in $X_9$; The time effect of the policy is insufficient, etc. At the same time, there is a more significant gap between acceptable policies and the above policies in other variables. P3, P4, and P12 do not explicitly mention the main responsibilities of any management body in $X_3$, resulting in the three policies scoring 0 on this variable. P7 and P12 are "interim" policies, but do not specify a temporary period or a time point for "conversion". P12 has even been "temporary" for ten years. The long-term "temporary" is not only unfavorable to the revision and improvement of the policy, but also restricts the implementation effectiveness of the policy.

**4. Unacceptable level policy.** The PMC-Index score of P13 is 4.34, which is rated as unacceptable. P13 was promulgated earlier, at that time, the relevant regulations and policies were not perfect, and the understanding of special fund management was still shallow. there is a large gap between P13 and recent policies in many dimensions.

In $X_4$, P13 does not explicitly stipulate the basic operational conditions such as the funding source, legal basis and budget arrangement of special funds, and the score of this variable is 0; In $X_5$, P13 does not indicate the specific management method of special funds, which is not in line with the current relevant regulations on special funds management; In $X_9$, P13 does not make corresponding requirements on the supervision subject, implementation time limit, asset ownership and performance evaluation of the special fund, which is not conducive to improving the efficiency of the use of the special fund.

## Discussion and conclusions

The provincial special public funding policy of China's private colleges and universities is the fundamental guide for local governments to use fiscal expenditure policy tools to support and standardize the development of private colleges and universities, and also the action guide for local governments to manage and use the special funds of private colleges and universities. The consistency and completeness of its policy texts directly affect the effectiveness of the government's public funding. Provincial policies are a comprehensive embodiment of following the overall requirements of the central government and based on local realities [49]. However, the policy orientation and external institutional environment of private higher education in different periods in China are different, and the quantity and quality of private higher education in different regions are still unbalanced [50]. As a result, the consistency and completeness of the text of the provincial special public funding policy for private colleges and universities have been different.

This study uses the PMC-Index model combined with text mining technology to quantitatively evaluate the policy texts of special public funding for private colleges and universities promulgated by 13 provinces in China, focusing on the analysis of the consistency, advantages and disadvantages of each policy, and the main conclusions can be summarized as follows. Firstly, the content design of the 13 policies was generally acceptable. The average PMC-Index of 13 policies is 6.97, and the average PMC-Surface map is smooth, which indicates that the policy consistency is relatively good. The internal development efforts in all aspects of the policy are balanced and the overall structure is reasonable. The PMC-Index of various policies showed an upward trend with the passage of publication time. Among them, there are 7 perfect or excellent policies, and only one unacceptable policy. Secondly, there is still room for improvement in the provincial special public funding policy of China's private colleges and universities. With the exception of $X_1$ (Policy type), the average PMC-Index score of the other 8 primary variables did not reach the maximum value. In particular, the average scores on $X_3$ (Responsibilities regulations), $X_6$ (Scope of assistance) and $X_7$ (Conditions of application) are much lower than other variables and need to be focused by policymakers. In addition, the "temporary" policy (P11, P12) that has not been revised for a long time has a significantly low PMC-Index score, poor consistency, and the content design needs to be strengthened.

There are some limitations in this study. Firstly, in the selection of policy samples, because most local governments adopt overall management of special funds for private colleges and universities, they do not focus on specific types and categories, and their management methods are relatively macro. Therefore, this article uses the special fund management measures issued by various places to represent the special public funding policy. However, strictly speaking, the management method is only a part of the special public funding policy system, and future research should further track the guiding policies issued by the state and the relevant implementation rules and funding arrangements issued by various localities, so as to achieve a quantitative evaluation of the special public funding policy system. Secondly, in terms of variable setting, although this paper comprehensively considers text mining, literature research and

expert guidance, the evaluation dimension and variable setting are still subjective. Future studies can use the grounded theory to gradually deduce more and deeper dimensions and variables, and establish a more scientific evaluation system. Thirdly, at the level of research methods, although the introduction of PMC-Index model in this paper can analyze the consistency, and advantages and disadvantages of policy contents, the analysis results can only show whether the policy samples contain relevant variables, and can not further evaluate the specific performance of these variables after the implementation of policies. Subsequent studies should focus on the evaluation system constructed in this paper to dynamically evaluate the specific performance of relevant variables in policy implementation.

## Author Contributions

**Conceptualization:** Maobo Hu.

**Funding acquisition:** Maobo Hu.

**Methodology:** Cai Guo.

**Resources:** Maobo Hu.

**Supervision:** Dan Ma.

**Validation:** Yang Wang.

**Visualization:** Yang Wang.

**Writing – original draft:** Cai Guo.

**Writing – review & editing:** Dan Ma.

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
