## [Decision Letter · Decision Letter 0]

3 Oct 2023

PONE-D-23-27566Quantitative evaluation of China’s private universities provincial public funding policies based on the PMC-Index modelPLOS ONE

Dear Dr. Guo,

Thank you for submitting your manuscript to PLOS ONE. After careful consideration, we feel that it has merit but does not fully meet PLOS ONE’s publication criteria as it currently stands. Therefore, we invite you to submit a revised version of the manuscript that addresses the points raised during the review process.

We look forward to receiving your revised manuscript.

Kind regards,

Muhammad Farooq Umer, PhD Epidemiology and Health Statistics

Academic Editor

PLOS ONE

Journal Requirements:

The funding was provided by National Social Science Fund (BIA190178) ‘Research on public funding model and performance evaluation of non-profit private colleges and universities’.

Reviewers' comments:

Reviewer's Responses to Questions

**Comments to the Author**

1. Is the manuscript technically sound, and do the data support the conclusions?

Reviewer #1: Yes

Reviewer #2: No

Reviewer #3: Yes

2. Has the statistical analysis been performed appropriately and rigorously? 

Reviewer #1: Yes

Reviewer #2: No

Reviewer #3: Yes

3. Have the authors made all data underlying the findings in their manuscript fully available?

Reviewer #1: No

Reviewer #2: No

Reviewer #3: Yes

4. Is the manuscript presented in an intelligible fashion and written in standard English?

Reviewer #1: Yes

Reviewer #2: No

Reviewer #3: Yes

5. Review Comments to the Author

Reviewer #1: This study uses PMC-index model and text mining technology to establish an evaluation index system for the provincial special public funding policy of private universities in China. The comments are as follows:

1. The authors mentioned that PMC-Index model has been used in some areas related to public policy. Has it been used in evaluating the funding policy in private education sector worldwide and specific countries? If yes, the authors need to address it and explain what this study can contribute to the relevant research area based on previous studies, e.g., improved research methods, new evidence.

2. There are some language errors need to be revised in the manuscript, e.g., P24 (Table 7), the Chinese characters should be revised to English.

3. Figures should be provided with high resolution, e.g., 300 dpi.

4. Some of the references are from journals in China. If the references are in Chinese language, the authors need to note the language name in each of the references.

Reviewer #2: It should be rejected because of limited originality.There are many very similar papers to your work in Chinese (some of them mentioned in your list of refs) and this renders your paper of limited novelty?

Reviewer #3: This manuscript uses PMC-index model and text 38 mining technology to establish an evaluation index system for the provincial special 39 public funding policy of private universities in China. These findings contribute to a comprehensive understanding of China's special public funding policies for private higher education institutions. The structure of the paper is logical and the research methodology and results are presented in a way that is clear and concise.

Advantages and benefits of the proposed approach should be given in detail. Also, the research gaps and the novelty of this study should be highlighted and summarized.

Methodology: The authors have used various research methods but need to justify why these were used over the others. For example: why PMC-Index model and not thematic analysis.

The methodology section requires a major revision. The authors should rewrite it professionally.

Provide a reference for the claim stated in lines 115 and 121.

The quality of Figure 1 is not good.

The 13 policies are not sufficiently representative of the overall situation and some of the relevant policies will not reflect the relevant terms in the headings. Therefore, in my opinion, the authors should consider increasing the sample size by enriching the search method for further studies.

Could you please check whether it is possible to additionally find articles published from 2022?

6. PLOS authors have the option to publish the peer review history of their article (what does this mean?). If published, this will include your full peer review and any attached files.

Reviewer #1: No

Reviewer #2: No

Reviewer #3: **Yes: **Hongxin Ma

---

## [Author Response · Author response to Decision Letter 0]

5 Nov 2023

Dear Editor, Dear reviewers

Thank you for your letter dated October 4. We thank the reviewers for the comments of the previous version of the manuscript. Their revision suggestions have helped us to improve our work. Based on the instructions provided in your letter, we uploaded the revised manuscript with all the changes highlighted in a different color (red).

Regarding the comments adopted for this study, we have made one-to-one corresponding revisions in the manuscript. 

If there is something inappropriate with our revisions or responses, please let us know at any time. We are pleasure to make further responses or amendments. 

Thank you again.

We hope that the revised manuscript be accepted.

Sincerely,

Cai Guo

We thank the editors and reviewers for comments and suggestions. Our response and revisions are as follows.

Reviewer #1:

Comment 1: The authors mentioned that PMC-Index model has been used in some areas related to public policy. Has it been used in evaluating the funding policy in private education sector worldwide and specific countries? If yes, the authors need to address it and explain what this study can contribute to the relevant research area based on previous studies, e.g., improved research methods, new evidence.

Response: Thank you for the comments and enlightening suggestions. The PMC-Index model has not been used used in evaluating the funding policy in private education sector worldwide and specific countries. According to your comments, we explain in the literature review why the PMC-Index model is suitable for evaluating funding policies in such private education sectors. We try to expand the scope of application of PMC-Index model, and use it to quantitatively evaluate the completeness and consistency of the content of the special financial assistance policy of China's provincial private universities, so as to provide a new basis for policy adjustment.

Comment 2-4: There are some language errors need to be revised in the manuscript, e.g., P24 (Table 7), the Chinese characters should be revised to English.Figures should be provided with high resolution, e.g., 300 dpi.Some of the references are from journals in China. If the references are in Chinese language, the authors need to note the language name in each of the references.

Response: Thanks for pointing out detailed errors in the syntax and formatting of the article. We have corrected these errors and provided high-resolution figures.

Reviewer #2:

Comment: It should be rejected because of limited originality.There are many very similar papers to your work in Chinese (some of them mentioned in your list of refs) and this renders your paper of limited novelty?

Response: Thank you for your comments. This paper has explained in the literature review that most of the evaluation studies on the public funding policies of China's private colleges and universities stay at the qualitative evaluation level of macro public funding policies, and lack of systematic evaluation of micro-specific public funding policies. At the same time, the existing quantitative evaluation studies on such policies mostly focus on the post-evaluation perspective, focusing on the effectiveness of the policy, and less quantitative analysis on the completeness and rationality of the policy content itself. Therefore, we believe that the innovation and novelty of this paper are mainly reflected in the following three aspects:

1.We focus on micro-specific funding policies, using text mining technology to dig deeper into the policy of provincial special public funding for private colleges and universities in China, and evaluate the specific content of the policy, which provides a new perspective for related research.

2.In this paper, the Policy Modeling Research Consistency (PMC) index model is applied for the first time to evaluate the funding policies of the private education sector in a specific country, and a quantitative evaluation framework is constructed for the text of the provincial special public funding policies of private colleges and universities in China, which provides a new method for the existing research.

3.This paper selects 13 policies as research samples and makes a quantitative evaluation of the advantages and disadvantages of each policy, which provides new evidence for the adjustment of provincial special public funding policies for private colleges and universities in China. 

Reviewer #3: 

Comment 1: Advantages and benefits of the proposed approach should be given in detail. Also, the research gaps and the novelty of this study should be highlighted and summarized. Methodology: The authors have used various research methods but need to justify why these were used over the others. For example: why PMC-Index model and not thematic analysis.The methodology section requires a major revision. The authors should rewrite it professionally.

Response: Thank you for the comments and enlightening suggestions. According to your comments, we have made the following modifications:

In the literature review section, we compare the PMC-Index model with other policy content evaluation methods, list the advantages of the PMC-index model in detail, and explain why this method was chosen in this paper. In the section on Samples and methods, we have added a section on "Research methods". In this section, we introduce and explain the various analysis methods used in this article. In addition, the novelty and limitations of this study are summarized in the introduction, discussion and conclusion.

Comment 2: Provide a reference for the claim stated in lines 115 and 121. The quality of Figure 1 is not good.

Response: Thanks for pointing out the normative error in the article. We have corrected these errors and provided high-resolution figures.

Comment 3: The 13 policies are not sufficiently representative of the overall situation and some of the relevant policies will not reflect the relevant terms in the headings. Therefore, in my opinion, the authors should consider increasing the sample size by enriching the search method for further studies.

Response: Thank you for your reminding and enlightening suggestion. Your suggestion is very pertinent. In the process of searching policy samples, we first focused on the research object, namely "provincial special public funding policies of private colleges and universities in China", and tried to retrieve the policy samples with the highest matching degree. However, after searching, we found that not all provinces have specially issued such policy texts, and we only retrieved 13 samples of current effective policies. The contents of these 13 samples are consistent in terms of hierarchy, structure and theme. In order to expand the sample size, we tried to search a wide range of policy texts involving the public funding of private universities. However, we found that in the content of these policies, the content and space devoted to public funding are less, and they are relatively macro. As a result, they are not well matched with the collected 13 policy samples in terms of hierarchy, structure and theme, and are not suitable for analysis by PMC-Index model. Therefore, we finally choose these 13 policies as samples and explain this defect in the last paragraph of the article.

Thank you again for this suggestion.

Comment 4: Could you please check whether it is possible to additionally find articles published from 2022?

Response:Thank you for your careful review. According to your comments, We added several articles published in 2022 or 2023.

---

## [Decision Letter · Decision Letter 1]

27 Nov 2023

Quantitative evaluation of China’s private universities provincial public funding policies based on the PMC-Index model

PONE-D-23-27566R1

Dear Dr. Guo,

We’re pleased to inform you that your manuscript has been judged scientifically suitable for publication and will be formally accepted for publication once it meets all outstanding technical requirements.

Kind regards,

Muhammad Farooq Umer, PhD Epidemiology and Health Statistics

Academic Editor

PLOS ONE

Additional Editor Comments (optional):

Reviewers' comments:

Reviewer's Responses to Questions

**Comments to the Author**

1. If the authors have adequately addressed your comments raised in a previous round of review and you feel that this manuscript is now acceptable for publication, you may indicate that here to bypass the “Comments to the Author” section, enter your conflict of interest statement in the “Confidential to Editor” section, and submit your "Accept" recommendation.

Reviewer #1: All comments have been addressed

Reviewer #3: All comments have been addressed

2. Is the manuscript technically sound, and do the data support the conclusions?

Reviewer #1: Yes

Reviewer #3: Yes

3. Has the statistical analysis been performed appropriately and rigorously? 

Reviewer #1: Yes

Reviewer #3: Yes

4. Have the authors made all data underlying the findings in their manuscript fully available?

Reviewer #1: Yes

Reviewer #3: Yes

5. Is the manuscript presented in an intelligible fashion and written in standard English?

Reviewer #1: Yes

Reviewer #3: Yes

6. Review Comments to the Author

Reviewer #1: The authors have replied the comments of the reviewer and haver revised the manuscript according to them. Thus, the current manuscript can be accepted after proofreading.

Reviewer #3: I think the references need to be updated substantially, as the current version has too many studies published from China (25), which is inappropriate for an international journal that caters for different research backgrounds and readers.

7. PLOS authors have the option to publish the peer review history of their article (what does this mean?). If published, this will include your full peer review and any attached files.

Reviewer #1: No

Reviewer #3: No

---

## [Editor Report · Acceptance letter]

30 Nov 2023

PONE-D-23-27566R1 

Quantitative evaluation of China’s private universities provincial public funding policies based on the PMC-Index model 

Dear Dr. Guo:

I'm pleased to inform you that your manuscript has been deemed suitable for publication in PLOS ONE. Congratulations! Your manuscript is now with our production department. 

Kind regards, 

on behalf of

Dr. Muhammad Farooq Umer 

Academic Editor

PLOS ONE